# Cellulose Synthesis in Cyanobacteria: Shared Pathways and Distinct Features with Bacteria and Plants

**DOI:** 10.3390/plants14172655

**Published:** 2025-08-26

**Authors:** Xinhui An, Vicente Ramírez, Markus Pauly

**Affiliations:** 1Institute for Plant Cell Biology and Biotechnology, Heinrich Heine University Düsseldorf, 40225 Düsseldorf, Germany; xinhui.an@hhu.de (X.A.); ramirezg@hhu.de (V.R.); 2Cluster of Excellence on Plant Sciences, 40225 Düsseldorf, Germany

**Keywords:** cyanobacteria, cellulose, cellulose synthase complex, bacteria, plants

## Abstract

Cellulose, the most abundant polymer on this planet, is widely produced by plants and many bacterial species. Certain cyanobacterial species also synthetize cellulose, though typically at much lower yields compared to other bacteria. Cyanobacteria are particularly intriguing in this context, as they uniquely combine the features of Gram-negative bacteria with plant-like features, such as oxygenic photosynthesis and CO_2_ fixation. This review highlights the structure and biosynthesis of cellulose in cyanobacteria, and explores the distinctive features compared with those of bacterial and vascular plants. We also discuss current strategies to enhance cellulose production in cyanobacteria through genetic engineering, synthetic redesign and environmental modulation, and propose key knowledge gaps. This review thus provides a foundation for advancing both fundamental understanding and the development of sustainable cellulose-based biotechnologies.

## 1. Introduction

Cellulose is the most abundant biopolymer on Earth and serves as a key structural component across a wide range of organisms. It is found embedded in the cell walls of terrestrial plants, within the tissues of algae, and in the epidermal membranes of tunicates [1,2]. In bacteria, cellulose acts primarily as a structural element of extracellular biofilms. It has also been identified as a component of the extracellular polysaccharide (EPS) matrix in certain cyanobacterial species, although typically produced in relatively low amounts [3]. Unlike bacterial cellulose, plant-based cellulose chains are held together by matrix polymers such as hemicelluloses and lignin to form a composite [4]. Due to its abundance plant-based cellulose has long supported industries such as paper, textile, packaging, and, more recently, biofuel production. In contrast, bacteria-based cellulose is synthesized in a highly pure form, free of other polymers, and exhibits a tunable nanoscale structure with high crystallinity. These characteristics have made it particularly attractive for biomedical applications, including wound dressings, drug delivery systems, and tissue engineering [4,5,6].

Cyanobacteria, or blue–green algae, are ancient Gram-negative, photosynthetic prokaryotes that originated approximately 3.5 billion years ago [7]. They display a wide range or morphological forms, from unicellular to multicellular structures, and exhibit remarkable ecological and physiological diversity. Cyanobacteria occupy a broad spectrum of habitats including freshwater and hypersaline environments, and have adapted to psychrophilic and thermophilic conditions [8]. Like plants, cyanobacteria perform oxygenic photosynthesis and are capable of fixing atmospheric CO_2_, making them promising candidates for carbon-negative biomanufacturing. The cellulose they produce shares key properties with bacterial cellulose, including a tunable nanoscale architecture and high purity due to the absence of lignin and hemicelluloses. Unlike heterotrophic bacteria, cyanobacteria use sunlight and CO_2_ as their sole energy and carbon sources, enabling sustainable production without the need for organic feedstocks. Additionally, their genetic tractability and relatively simple metabolic networks make them ideal platforms for synthetic biology applications. Scaling up cyanobacterial cellulose production could open new avenues for producing high-value nanocellulose materials with applications in medicine, electronics, and environmental engineering [9].

## 2. Cellulose Structure

Cellulose is a crystalline polymer composed of β-1,4-linked glucose units [10,11]. Structurally, it adopts a two-fold helical conformation formed by repeating cellobiose units, a result of the 180° rotation imposed by β-1,4 glycosidic linkages between consecutive glucose molecules [12]. Four distinct crystalline allomorphs of cellulose—designated cellulose I through IV—have been identified, with cellulose I and II being the most prevalent [13,14]. These forms can be distinguished by their hydrogen bonding patterns using X-ray diffraction and other spectroscopic techniques [15].

Cellulose I, produced almost exclusively in nature [10,16], can be irreversibly converted to cellulose II through strong alkali treatment, indicating that cellulose I is metastable while cellulose II is the thermodynamically stable form [17]. Cellulose I is composed of laterally and unidirectionally aligned parallel β-1,4-glucan chains. Morphologically, it appears as submicroscopic rods known as microfibrils [17]. These microfibrils are typically a composite of two sub-allomorphs: cellulose Iα and cellulose Iβ [18]. Cellulose Iα has a triclinic crystal structure consisting of a single molecular chain, whereas cellulose Iβ features a monoclinic structure comprising two parallel chains [10,19]. Cellulose Iα is enriched in bacterial and algal cellulose, while higher plants predominantly synthesize cellulose Iβ [20,21,22,23]. However, exceptions exist, such as the nearly pure cellulose Iα observed in the alga *Glaucocystis* [24]. Cyanobacteria predominantly produce cellulose I (Table 1). This has been confirmed in several strains, including *Oscillatoria* sp. UTEX L2435, *Nostoc* sp. UTEX 2209, *Gloeocapsa* sp. UTEX L795, *Scytonema hofmanni* UTEX 2349, *Anabaena* sp. UTEX 2576, and *Phormidium autumnale* UTEX 1580 [3].

In contrast to cellulose I, cellulose II is arranged in an antiparallel manner of glucan chains due to the extensive intersheet hydrogen bonding [15,25]. When compared with cellulose I, cellulose II is more chemically reactive, has a lower molecular weight, and possesses a greater affinity for dyes [17,26]. Cellulose II is only known to be synthesized naturally by the marine alga *Halicystis* [27], the Gram-positive bacterium *Sarcina ventriculi* [28], and by the mutants and agitated cultures of *Komagataeibacter xylinus* (*K. xylinus,* also known as *Acetobacter xylinum* or *Gluconacetobacter xylinus*) [29] and Cyanobacteria *Nostoc punctiforme* ATCC 29133 (Table 1) [3].

## 3. Occurrence and Function of Cellulose

Cyanobacteria generally exhibit a biochemical composition characterized by relatively low carbohydrate content, ranging from 10–30% under optimal growth conditions, significantly lower than 70–80% carbohydrate content observed in vascular plants [30,31]. In cyanobacteria, carbon is primarily stored as intracellular glycogen or secreted as extracellular polysaccharides (EPSs), which are categorized into three types: sheaths (thin, dense layers around cells), capsules (thick, structured layers closely attached to cells), and slimes (loose, gelatinous coatings that do not match the cell shape) [32,33]. In contrast, plants store carbohydrates as intracellular starch but allocate a significant portion of their fixed carbon to the construction of cellulose-rich cell walls [34].

Unlike plants, where cellulose is a universal and essential component of both primary and secondary cell walls, the presence and production of cellulose in cyanobacteria is not ubiquitous and rather species-specific. Some species such as *Synechocystis* sp. PCC 6803, lack detectable levels of cellulose [35,36], whereas others like *Synechococcus* sp. PCC 7002 naturally synthesize cellulose forming a laminated layer between the plasma and outer membranes [15]. In several strains, cellulose is also found within the extracellular matrix as a minor component of the EPSs (Table 1) [3]. Notable examples include cellulose in the EPS of *Atacama* LLC-10 [37], within the sheaths of *Gloeocapsa* sp. UTEX L795, *Anabaena* sp. UTEX 2576, *Nostoc* sp. UTEX 2209, and *Scytonema hofmanni* UTEX 2349, as well as in the slime tube of *Oscillatoria* sp. UTEX L2435 and *Phormidium autumnale* UTEX 1580 [3].

The specific role of cellulose in cyanobacteria remains poorly understood. It is generally considered a minor component of EPSs, which perform numerous essential functions in cyanobacterial physiology and ecology. EPSs form a protective barrier around cells, shielding them from environmental stresses such as desiccation, heat, ultraviolet radiation, high salinity, predation, infection, and other external threats [38,39,40,41]. In motile cyanobacteria, EPSs also contribute to gliding motility by facilitating slime extrusion through junctional pore complexes [42], particularly during the hormogonia stage. Furthermore, EPSs mediate adhesion to surfaces through a cohesive three-dimensional polymer matrix, enabling cell–cell communication (either as vehicles of compounds or as quorum sensing molecules), and retention of extracellular enzymes [33]. EPSs can also be involved in biofilm formation, having a major role in shaping the complex microbial community structure of mats [43,44]. Under conditions of carbon excess, particularly when the carbon-to-nitrogen ratio is imbalanced, EPSs can also act as a carbon sink, storing surplus fixed carbon [45]. To what extent cellulose in the EPS contributes to these functions is not known.

## 4. Molecular Mechanisms of Cellulose Biosynthesis

Cellulose biosynthesis is well characterized in both bacterial and plant systems, where it is catalyzed by membrane-embedded cellulose synthase complexes (CSCs) that polymerize β-1,4-glucan chains from cytosolic UDP-glucose. In bacteria, four major types of cellulose synthase (*bcs*) operons have been identified, classified as type I–IV [46]. These types differ in gene composition, operon organization, and accessory proteins.

Type I operons, first described in *K. xylinus*, typically consist of the *bcsABCD* gene cluster, where BcsA and BcsB form the catalytic core responsible for glucan formation, while BcsC and BcsD facilitate export and microfibril assembly. Type II operons, found in organisms such as *Escherichia coli* and *Salmonella enterica*, include additional regulatory genes such as *bcsE* and *bcsG* but lack *bcsD*. Type III operons, present in *Agrobacterium tumefaciens*, are more variable. They often lack *bcsC* while retaining *bcsD* and include unique accessory genes with largely unknown functions. Type IV operons are present in cyanobacteria such as *Nostoc* sp. PCC 7120 and *Thermosynechococcus vulcanus* (*T. vulcanus*). These operons typically entail *bcsA* but lack *bcsB* and other canonical *bcs* genes present in other bacterial systems. Certain cyanobacteria utilize alternative genes for glucan translocation and assembly. For instance, genes encoding HlyD-like membrane fusion proteins and the cyanobacteria-specific protein BcsW appear to be functionally equivalent to BcsB, enabling the export of nascent cellulose chains. These genes are often located within the same operon as *bcsA*, suggesting a coordinated regulatory and functional role in cellulose biosynthesis [46].

### 4.1. Cellulose Synthase Genes

Despite their vast evolutionary divergence, bacterial cellulose synthase A (BcsA, Figure 1a), cyanobacterial extracellular cellulose synthase A (XcsA; Figure 1b), and plant cellulose synthase A (CesA, Figure 1c) share key topological features. All three are integral membrane proteins with multiple transmembrane (TM) segments that form a channel for cellulose extrusion across the plasma membrane. They also share a conserved catalytic core, where the central glycosyltransferase domain contains an invariant sequence motif: D,D,D,QXXRW. This motif is critical for catalytic activity and is conserved not only at the amino acid sequence, but also in the tertiary structure, highlighting a common enzymatic mechanism across bacteria, cyanobacteria and plants [46,47,48]. This domain can be split into four subdomains, U1–-U4, in which the first three subdomains each contain one aspartic acid motif and the last subdomain contains the Q(Q/R)XRW motif in which X can be any amino acid [49].

Reflecting adaptations to their respective cellular environments and regulatory systems, bacterial BcsA, cyanobacterial XcsA and plant CesA exhibit significant differences in overall sequence and domain organization. Cyanobacterial XcsA is likely the evolutionary ancestor of plant CesA and occupies a unique phylogenetic and structural position that bridges the bacterial and plant systems, yet remains comparatively less well understood mechanistically. Some cyanobacterial variants more closely resemble bacterial BcsA, while others share greater similarity with plant CesA.

Bacterial BcsA comprises eight TMs—four near the N-terminus and four near the C-terminus—that form a tunnel for cellulose translocation, and two large cytoplasmic domains (Figure 1a). These include a central glycosyltransferase domain with four conserved subdomains (U1–U4) and a C-terminal PilZ domain. The PilZ domain plays a key regulatory role by specifically binding the bacterial second messenger bis-(3′,5′)-cyclic di-guanosine monophosphate (c-di-GMP), which activates BcsA to polymerize UDP-glucose into cellulose. Intracellular c-di-GMP levels, governed by diguanylate cyclases and phosphodiesterases, thus precisely control cellulose biosynthesis in response to environmental cues [46].

Plant CesA retains the central glycosyltransferase domain and is similarly embedded in the membrane via eight TM helices, two near the N-terminus and six near the C-terminus, also forming a translocation tunnel (Figure 1c) [50,51]. However, it lacks the PilZ domain and is regulated through distinct plant-specific mechanisms. Notably, plant CesA contains an N-terminal RING-like zinc-finger domain (NTD), which is involved in protein–protein interactions and is crucial for dimerization or oligomerization [52]. Additionally, two conserved regions within the glycosyltransferase domain—the plant-conserved region (PCR) between U1 and U2 and the class-specific region (CSR) between U2 and U3 subdomains—are thought to mediate assembly of CesA into a characteristic six-lobed rosette cellulose synthase complex unique to plants [48].

Such differences in domain architecture between CesA and BscA have functional consequences. In bacteria, BcsA regulation by c-di-GMP provides a rapid and direct mechanism to control cellulose biosynthesis. In contrast, plant CesA relies on the precise assembly of the rosette complex to achieve proper cellulose crystallinity and deposition. Disruptions in these domains can significantly impair function: for instance, mutations in the catalytic or TM regions of *Arabidopsis* CesAs reduce cellulose crystallinity [53].

In cyanobacteria, two distinct structural types of extracellular cellulose synthases have been identified (Table 1) [54]. One structural type mirrors the typical domain architecture of bacterial BcsA, comprising eight TM helices and a PilZ domain, and is referred to as a “cellulose synthase with a PilZ domain” (Figure 1b, left panel). This type is widespread among cyanobacterial species, including and *Synechococcus elongatus* PCC 7942 *(ABB57428.1,* also known as *Synechococcus leopoliensis* strain UTCC 100). The second structural type features a modified domain structure: presenting only seven TM helices, two near the N-terminus and five near the C-terminus, the absence of a C-terminal PilZ domain, and the addition of a plant-like PCR domain between the U1 and U2 subdomains, and is referred to as “cellulose synthase with a PCR domain” (Figure 1b, right panel). This structure is found in some cyanobacterial species, particularly in diazotrophic cyanobacteria such as *Nostoc* sp. PCC 7120 (*NP_487797.1*) and *Nostoc punctiforme* ATCC 29133 (also known as *Nostoc punctiforme* PCC 73102, *Npun_F1469*), and *Synechococcus* sp. PCC 7002 (*SYNPCC7002_A2118*, Figure 1b, right panel) [3,54,55,56]. These structural variations suggest evolutionary transitions in domain organization and regulation, offering insights into the diversification of cellulose synthases across cyanobacterial lineages [55].

**Figure 1 plants-14-02655-f001:**
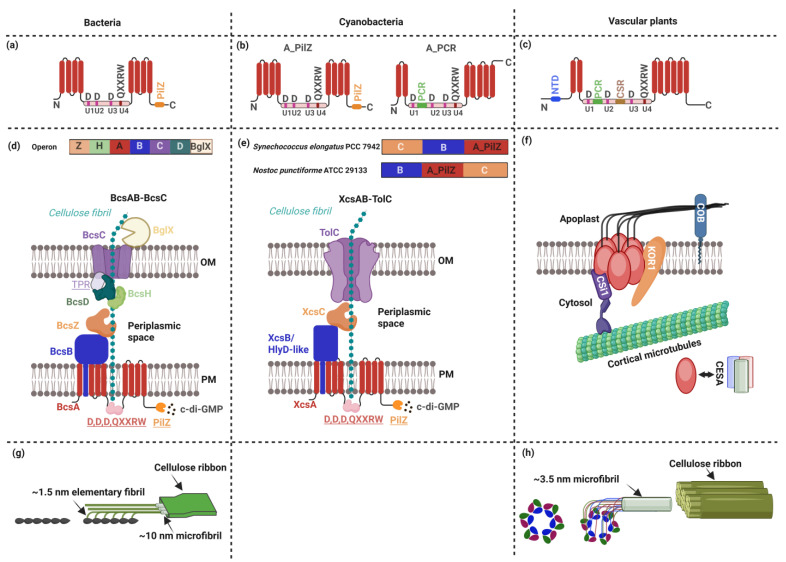
Comparative architecture and functional organization of cellulose synthase systems in bacteria, cyanobacteria, and plants. (**a**–**c**) Membrane topology of cellulose synthase A proteins, highlighting key functional domains and transmembrane segments. (**a**) Bacterial BcsA; (**b**) cyanobacterial XcsA; (**c**) plant CesA. Red bars indicate transmembrane regions; pink segments indicate glycolsyltransferase domains. N, N terminal; C, C terminal; D, catalytic domain; QXXRW, U1, U2, U3, U4, PilZ, PCR, NTD and CSR domains are also indicated. In panel (**b**), A_PilZ refers to *XcsA* variants with a PilZ domain; A_PCR refers to *XcsA* variants with a PCR domain. (**a**–**c**) Adapted from Milou Schuurmans et al. (2014), and Noack and Persson (2023) [48,55]. (**d**–**f**) Comparative models of cellulose synthase complex (CSC) in representative (**d**) bacteria (BcsAB-BcsC), (**e**) cyanobacteria (XcsAB-TolC), and (**f**) plants (CESAs). Gene products are color-coded consistently with the panel. Protein domains are underlined. OM, out membrane; PM, plasma membrane. (**d**) The letters A, B, C, D, H, Z, and BglX typically denote the genes *bcsA*, *bcsB*, *bcsC*, *bcsD*, *bcsH*, *bcsZ* and *BglX,* respectively. (**e**) Genes labeled A, B, and C in *Synechococcus elongatus* PCC 7942 and *Nostoc punctiforme* ATCC 29133 encode a cellulose synthase with a PilZ domain, an HlyD-like protein and an endoglucanase, respectively. In XcsAB-TolC model, XcsA (*Tll0007*, with a PilZ domain), XcsB/HlyD-like protein gene (*Tlr0903*), XcsC (*Tlr1902*). (**f**) In plants, CSCs form rosettes, each composed of six lobes, with three CesA subunits per lobe, totaling 18 CesAs per CSC. The accessory proteins CSl1, KOR1, and COB are also indicated. (**d**–**f**) Adapted from Maeda et al. (2018), Speicher et al. (2018), and Krasteva (2024) [56,57,58]. (**g**,**h**) Relationship between CSC membrane organization and cellulose microfibril structure in bacteria (possibility cyanobacteria) and plants. (**g**) In bacteria (and possibility in cyanobacteria), cellulose synthase complexes produce ~1.5 nm elementary fibrils that align into linear arrays, which further assemble into cellulose ribbons. (**h**) In plants, cellulose synthases form C3 trimeric lobes that 212 assemble into C6 rosettes, which produce ~3.5 nm cellulose microfibrils. Not drawn to scale. (g–h) Adapted from Wilson et al. (2021) [59]. All illustrations were created using BioRender.

**Table 1 plants-14-02655-t001:** Summary of cellulose occurrence and type, and structural types of cellulose synthases in cyanobacteria (N.D.: not detected; NA: data not available).

Strain	Strain Synonym	Cellulose		Cellulose Synthase A		Reference
		Occurrence	Type	Protein	Structure Type	
*Anabaena* sp. UTEX 2576	NA	Sheaths of EPS	I	NA	NA	[3]
*Atacama* LLC-10	NA	EPS	NA	NA	NA	[37]
*Crinalium epipsammum* ATCC 49662	NA	NA	NA	NA	NA	[3]
*Gloeocapsa* sp. UTEX L795	NA	Sheaths of EPS	I	NA	NA	[3]
*Nostoc punctiforme* ATCC 29133	*Nostoc punctiforme* PCC 73102	NA	II	Npun_F1469	with PCR domain	[3,56]
*Nostoc punctiforme* ATCC 29133	*Nostoc punctiforme* PCC 73102	NA	II	Npun_F6500	with PilZ domain	[3,56]
*Nostoc* sp. PCC 7120	*Anabaena* sp. PCC 7120	NA	NA	NP_487797.1, alr3757	with PCR domain	[3,54,56]
*Nostoc* sp. UTEX 2209	*Nostoc muscorum* UTEX 2209	Sheaths of EPS	I	NA	NA	[3]
*Oscillatoria* sp. UTEX L2435	NA	Slime of EPS	I	NA	NA	[3]
*Phormidium autumnale* UTEX 1580	NA	Slime tube/sheath	I	NA	NA	[3]
*Scytonema hofmanni* UTEX 2349	NA	Sheaths of EPS	I	NA	NA	[3]
*Synechococcus elongatus* PCC 7942	*Synechococcus leopoliensis strain* UTCC 100	NA	NA	ABB57428.1, SynPCC7942_1398	with PilZ domain	[3,56]
*Synechococcus elongatus* PCC 7942	*Synechococcus leopoliensis strain* UTCC 100	NA	NA	SynPCC7942_2151	with PilZ domain	[3,56]
*Synechococcus* sp. PCC 7002	NA	Layer between peptidoglycan / out membrane	I	SynPCC7002_A2118	with PCR domain	[3,15,54,56]
*Synechocystis* sp. PCC 6803	*Synechocystis* sp. ATCC 27184	N.D.	NA	NA	NA	[35,36]
*Thermosynechococcus elongatus* BP-2	NA	NA	NA	NP_680798	with PilZ domain	[54]
*Thermosynechococcus vulcanus*	NA	NA	NA	tll0007	with PilZ domain	[56]
*Thermosynechococcus vulcanus* RKN	NA	NA	NA	BAJ61012.1	with PilZ domain	[56]

### 4.2. Cellulose Synthase Complex

Cellulose synthase complexes (CSCs)**,** initially identified in *K. xylinus* bacteria and later recognized across diverse organisms, including bacteria, cyanobacteria, higher plants, and tunicates, play a central role in cellulose biosynthesis [48,60,61]. While these complexes are conserved in function, their structural organization and associated components vary significantly across taxa.

In bacteria, the genes encoding the CSCs are organized in a linear operon structure centered around the core cellulose synthase subunit *bcsA* (Figure 1d, upper panel). For example, *bcsA*, *bcsB*, *bcsC*, *bcsD*, *bcsH*, *bcsZ,* and *bglX* are often found clustered together, facilitating the coordinated transcription and regulation of cellulose biosynthesis. At the protein level, BcsB (Figure 1d, lower panel), which resides in the periplasm and is anchored to the plasma membrane via a single transmembrane helix, partners with BcsA to form the BcsAB complex. This complex forms a continuous channel that begins at the glycosyltransferase domain of BcsA, spans the membrane, and opens into the periplasm, enabling synchronized polymerization and translocation of glucan chains [62]. Cellulose is subsequently exported through the outer membrane via BcsC, a β-barrel protein with periplasmic tetratricopeptide repeats (TPRs), which is essential for cellulose secretion in Gram-negative bacteria [63]. Additionally, cellulose-complementing protein A (CcpAx, also known as BcsH) is hypothesized to be involved in the structural organization of the terminal complexes, cooperating with BcsD [64]. Other components include the endoglucanase BcsZ (also known as CmcAx), and bglX, a β-glucosidase, that appear to assist cellulose biosynthesis by hydrolyzing tangled glucan chains, when a failure in chain arrangement occurs [65,66]. Disruption of *bcsZ* in *K. xylinus* led to reduced cellulose production and irregular fibril architecture [65]. Similarly, *bcsZ* mutants in *Rhizobium leguminosarum* produced longer cellulose microfibrils but showed impaired biofilm formation [65,67]. For further information refer to Römling and Galperin (2015) and Buldum and Mantalaris (2021) [46,61].

In cyanobacteria, a similar functional framework appears to exist, although with notable genomic and structural distinctions. Previous studies have revealed that the genes encoding components of this complex often exhibit synteny, a conserved gene order, across various cyanobacterial species (Figure 1e, upper panel) [56]. For instance, in both *Synechococcus elongatus* PCC 7942 and *Nostoc punctiforme* ATCC 29133, the genes encoding the XcsA homolog (a cellulose synthase with a PilZ domain; *SynPCC7942_1398*, *Npun_F6500*), the XcsB homolog (an HlyD-like membrane fusion protein; *SynPCC7942_1399*, *Npun_F6499*), and XcsC homolog (an endoglucanase; *SynPCC7942_1400*, *Npun_F6501*) are organized within a conserved operon structure, resembling the gene arrangement observed in Gram-negative bacteria. In contrast, this syntenic arrangement is disrupted in *T. vulcanus*, where the corresponding genes are scattered across the genome: *tll0007* (encoding XcsA, a cellulose synthase with a PilZ domain), *tlr0903* (encoding XcsB, an HlyD-like protein), and *tlr1902* (encoding XcsC, an endoglucanase) are not co-located. Despite this genomic separation, homology modeling studies suggest a tripartite secretion system XcsAB-TolC analogous to the bacterial BcsAB–BcsC complex (Figure 1e, lower panel) [56].According to this model, XcsA (*tll0007*) in the plasma membrane is functionally connected to the outer membrane TolC-like channel *(tlr1605)* via a periplasmic adaptor protein, XcsB (*tlr0903*). Despite this tripartite arrangement providing a promising structural framework, its validity remains to be experimentally confirmed.

In contrast to the linear assemblies found in bacteria and cyanobacteria, higher plants possess unique rosette-shaped CSCs with six-fold radial symmetry (Figure 1f) [10,29,68,69]. These rosette complexes are composed of multiple non-identical cellulose synthase (CESA) isoforms, each playing specialized roles. CESAs assemble in Golgi and associate with various accessory subunits that regulate their trafficking and activity [10,48,59]. For instance, the Golgi-localized STELLO1 and STELLO2 (STL1/2) proteins directly interact with CESA subunits, and modulate CSC dynamics at the plasma membrane by facilitating proper complex assembly during intracellular trafficking and influence CSC dynamics [70]. They are functionally analogous to bacterial BcsH in organizing synthase complexes. Additionally, putative membrane-spanning endo-1,4-β-D-glucanase, KORRIGAN (KOR1) contributed to cellulose crystallinity [71,72], CSC velocity [73], and microfibril orientation [74], analogously to BcsZ’s role in bacteria and XcsC’s role in cyanobacteria. CSCs are packaged into vesicles and then pass through the Golgi and trans-Golgi network [75,76], which are guided to defined membrane sites aligned with cortical microtubules. These microtubule sites are marked by cellulose synthase interactive protein 1 (CSI1/POM2), which acts as a molecular landmark [77]. In contrast, SHOU4 and SHOU4-LIKE (SHOU4/4L) proteins interact with CESA proteins and function as negative regulators of CSC exocytosis. SHOU4 and SHOU4-LIKE (SHOU4/4L) proteins interact with CESAs to form transient inhibitory complexes, limiting the number of CSCs that reach the plasma membrane, and thereby modulating CESA trafficking. Another important factor, the glycosyl–phosphatidyl inositol-anchored protein, COBRA (COB) evolved alongside CESA, coinciding with the shift in linear arrays to rosette-shaped CSCs [78], therefore may be important in synthesizing glucan chains in close proximity to one another. For additional information, refer to [10,59,79].

Although the rosette-shaped cellulose synthase complexes (CSCs) in plants are structurally distinct from the linear assemblies observed in bacteria and cyanobacteria, many of their associated regulatory and accessory proteins exhibit striking functional analogies. These parallels may offer valuable insights into the identification of the currently uncharacterized components and mechanisms involved in cyanobacterial cellulose biosynthesis.

### 4.3. Organization of Cellulose Microfibrils

Traditional high-resolution imaging techniques, such as electron microscopy, have been extensively used to measure cellulose microfibrils in diverse taxa. In bacteria, CSCs are organized into extended rows along the cell membrane (Figure 1g). Each CSC produces an elementary fibril, and these fibrils further associate along the row to form thicker microfibrils. These microfibrils bundle into larger structures known as cellulose “ribbons” [25,59]. The assembly of twisted-ribbon structures has been observed in *K. xylinus*, which is prevented by the presence of carboxymethylcellulase (*Cmcax*) [65]. Microfibril assembly is believed to occur through a stepwise, hierarchical process [25]. The diameter of bacterial cellulose fibrils varies depending on their structural level and the bacterial species involved. In *K. xylinus*, each elementary fibril is approximately 1.5 nm in diameter and assembles into larger fibrils with an average width of 45.8 ± 2.0 nm and an average length of 906.1 ± 37.4 nm (n = 34) [65]. In contrast, cellulose fibrils produced by *Rhizobium leguminosarum* range from 5 to 6 nm in diameter and can extend up to 10 μm in length [80].

In cyanobacteria, the mechanism by which CSC arranges microfibrils remains unknown and might vary across different species or even strains, reflecting the distinct cellulose microfibril morphologies observed. For example, cellulose from *N. muscorum* UTEX 2209 aggregates into twisted ribbon-like structures reminiscent of the cellulose ribbons of *K. xylinus*, but in *C. epipsammum* ATCC 49662, cellulose microfibrils, appear to be very thin in both dimensions in comparison with the other cyanobacteria tested. While cellulose microfibrils in *P. autumnale* UTEX 1580 appear more discontinuous and irregular, in *Nostoc punctiforme* ATCC 29133, they are shorter and many have tapered ends, suggesting the presence of cellulose II. Their thickness ranged from 1.1 to 2.8 nm, with an average of 1.7 ± 0.4 nm, while the width was more variable and ranged from 5 nm to over 17 nm, with an average of 10.3 ± 4.1 nm [3].

In plants, each cellulose synthase (CESA) protein is proposed to produce a single glucan chain [81,82]. Plant rosette CSCs typically synthesize microfibrils composed of either 18 or 24 glucan chains [10]. The widely accepted model for Arabidopsis primary cell walls proposes an 18-chain CSC formed by six lobes, each containing the heterotrimers of three essential CESA isoforms (Figure 1h) [83,84]. In contrast, larger 24-chain CSCs are found in celery primary walls, gymnosperm secondary cell walls, and wood, consistent with their thicker microfibrils [10,85,86]. Rosette CSCs typically produce microfibrils with small diameters ranging from 2 to 3.5 nm. However, in some cases, single or multiple rows of terminal complexes can generate much larger structures, either thicker microfibrils up to 25 nm in diameter or flat cellulose ribbons reaching widths of up to 100 nm [48].

Taken together, although cellulose is present in cyanobacteria, its organization remains unclear. These well-characterized models in bacteria and plants offer valuable comparative frameworks for advancing our understanding of cyanobacterial cellulose organization, which remains a relatively underexplored frontier.

## 5. Enhanced Cellulose Production in Cyanobacteria

Despite the growing interest in cyanobacteria as sustainable biofactories, the native production of cellulose in most cyanobacterial species remains very low (e.g., *Synechococcus elongatus* PCC 7942: approximately 0.08 mg·mL^−1^·OD_750_^−1^) compared with bacteria (e.g., *K. xylinus*: 3.5–15.3 g·L^−1^ under various culture conditions) [87] and plants (e.g., reed, trees, cotton: approximately 40% or more of plant biomass) [88]. This limited yield presents a major barrier to realizing the full potential of cyano-cellulose in industrial applications.

To address the low productivity of natural strains, several genetic strategies have been developed to engineer cyanobacteria with enhanced cellulose biosynthesis capacity. Nobles and Brown (2008) functionally expressed a partial cellulose synthase operon *acsABΔC* (referred to *bcsABΔC*) from *K. xylinus* strain ATCC 53582 in the cyanobacterium *Synechococcus elongatus* PCC 7942, resulting in a significant increase in total cellulose-derived glucose content from 0.08 to 0.26 mg·ml^−1^ OD_750_^−1^ [89]. Additionally, Su et al. (2010) transferred cellulose synthesis genes (*bcsAB*) from *K. xylinus* ATCC 23769 to the heterocystous cyanobacterium *Nostoc* sp. strain PCC 7120 [90] and *Synechococcus elongatus* PCC 7942 [91], achieving even higher gains in cellulose-derived glucose compared to wild-type strains: 0.436–0.553 vs. 0.16 mg mL^−1^·OD_750_^−1^ for *Nostoc* sp. strain PCC 7120, and 0.325 vs. 0.078 mg mL^−1^·OD_750_^−1^ for *Synechococcus elongatus* PCC 7942. These results highlight the potential of heterologous gene expression in overcoming inherent limitations in native cellulose biosynthesis pathways.

Further progress was made with *Synechococcus* sp. PCC 7002, which naturally forms a thin cellulose layer localized between the peptidoglycan (PG) layer and the outer membrane. Seven genes from *K. xylinus*—*cmc*, *ccp*, *bcsA*, *bcsB*, *bcsC*, *bcsD*, and *bgl*—were heterologously expressed in *Synechococcus* sp. PCC 7002, but enhanced cellulose yields were only observed when the native *cesA* gene (responsible for endogenous cellulose layer formation in *Synechococcus* sp. PCC 7002, *cesA* with PCR domain) was inactivated and cells were cultured at low salinity. Notably, the partial expression of this operon, such as constructs containing only *bcsA*, *bcsB*, *bcsC,* and *bcsD*, failed to produce detectable extracellular cellulose, indicating that complete operon integrity is essential for functional cellulose biosynthesis. Among these, *cmc* and *ccp* likely assist in polymerization and fiber structuring, while *bcsD* and *bgl* contribute to cellulose crystallization and export. Representing a substantial improvement in cellulose quality and yield, the fully engineered strain produced cellulose amounting to around 14% of the cell dry weight within 12 days with approximately 65% type I crystalline structure [15].

In addition to genetic engineering, abiotic factors such as temperature and light have been shown to significantly influence cyanobacterial cellulose production. Kawano et al. were the first to demonstrate a light- and low-temperature–induced cell aggregation in the thermophilic cyanobacterium *T. vulcanus* is driven by the accumulation of extracellular cellulose. In their study, cells cultivated under light at 31 °C (low temperature) exhibited markedly enhanced aggregation and produced approximately twice as much cellulose as those grown at the normal temperature of 45 °C. This increase was attributed to the cellulose synthase gene with a PilZ domain *Tll0007*, which accounted for roughly 50% of total cellulose production under these stress conditions [92]. In addition, the authors identified a regulatory pathway involving the blue-light receptor SesA and two accessory photoreceptors, SesB and SesC, which modulate intracellular c-di-GMP levels in response to light. The increased c-di-GMP activates the catalytic subunit of cellulose synthase, *Tll0007*, through binding to its PilZ domain, ultimately leading to enhanced cellulose synthesis and cell aggregation [56]. These findings suggest that finetuning environmental conditions such as temperature and light can help enhancing the production of extracellular cellulose in thermophilic cyanobacteria.

## 6. Key Research Gaps in Understanding and Engineering Cyano-Cellulose Pathways

Despite recent advances in characterizing cyanobacterial cellulose biosynthesis, many aspects of the underlying mechanisms remain unresolved.

The functional characterization of the two types of cellulose synthases found in cyanobacterial remains incomplete. For instance, the cellulose synthase in *Nostoc* sp. PCC 7120 (*NP_487797.1*; WP_010997898.1) lacks the canonical PilZ domain but contains a PCR domain. Whether this variant operates through a mechanism homologous to plant CesAs or represents a case of convergent evolution remains to be determined. Structural and biochemical studies are needed to elucidate its catalytic function and interaction partners. Another unresolved issue is whether multiple independent cellulose synthase systems coexist within a single cyanobacterial genome. For instance, *Synechococcus elongatus* PCC 7942 contains two putative cellulose synthase operons: one containing *Synpcc7942_1398* (a cellulose synthase gene with a PilZ domain) and *Synpcc7942_1399* (an HlyD-like gene); the second including *Synpcc7942_2151* (a cellulose synthase gene with a PilZ domain) and *Synpcc7942_2152* (an HlyD-like gene). Similarly, *Nostoc punctiforme* ATCC 29133 contains two distinct cellulose synthase genes, *Npun_F1469* with a PCR domain and *Npun_F6500* with a PilZ domain. It remains unclear whether these represent two distinct cellulose synthase systems with different biological roles, such as wall synthesis versus biofilm formation, or if one operon encodes regulatory or redundant components, or if they are simply evolutionary remnants of hybrid system. At the level of the terminal cellulose synthase complex, the current model of cyanobacterial cellulose synthesis primarily considers XcsC as an endoglucanase. However, it remains unclear whether additional regulatory or accessory proteins are involved in the assembly and/or stabilization of cellulose fibers. Additionally, the secretion mechanism of cellulose in cyanobacteria remains poorly defined. Although a model involving the XcsAB-TolC pathway has been proposed, it lacks experimental validation. In this model, the proteins XcsA, XcsB, and TolC are not encoded within the same operon, raising questions about the pathway’s functional coherence. Whether similar mechanisms apply to operon-organized gene clusters remains an open question and warrants further investigation. Moreover, at the level of cellulose microfibril organization, it is not yet known how different cyanobacterial species align and deposit cellulose fibers. Whether they align them into ordered structures similar to those observed in bacterial or plant systems is still an open question. From both biological and biotechnological perspectives, engineering of the PilZ domain or modulating c-di-GMP levels, and optimizing cultivation conditions, such as light intensity, nutrient availability, and stress induction, offer interesting opportunities to significantly enhance cellulose yield. However, these approaches could benefit from a deeper understanding of the underlying molecular basis of the mechanisms involved. Also, significant challenges remain in the harvesting, purification, and scale-up steps involved in the industrial production of cyanobacterial cellulose.

Addressing these research gaps will not only deepen our understanding of cellulose biosynthesis in cyanobacteria, but also unlock new strategies in sustainable biomaterials and engineered microbial systems.

## Data Availability

Data sharing is not applicable.

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
