# Peer review of "Cellulose Synthesis in Cyanobacteria: Shared Pathways and Distinct Features with Bacteria and Plants"

_plants, 2025, doi:10.3390/plants14172655_

Round 1

Reviewer 1 Report

Comments and Suggestions for Authors

This is an interesting and well-written review comparing cellulose synthesis in cyanobacteria, plants and bacteria, highlighting the gaps in our understanding of the molecular mechanisms behind cellulose synthesis in cyanobacteria and opening new axes for further research.

I would suggest, if possible, to include a table summarizing the type of cellulose (I or II), where cellulose is detected, type of cellulose synthase XcsA and microfibrils arrangement for the cyanobacteria species cited in the manuscript.

For paragraph 5 (Enhanced cellulose production in cyanobacteria), it would be interesting to put the number given for the cellulose production of cyanobacteria after genetic engineering in perspective with production from bacteria or extraction from plant tissue.

Reviewer 2 Report

Comments and Suggestions for Authors

-38-39, please add reference

-140-141, authors should add the species of these genes

-fig 1f, please recheck if the synthesis is for 1 single fiber as in d and e and those fibers assemble to microfiber? or it is really produce as microfiber at once.

-227, please add shortly k.xylinus is a bacteria

-327, based on this please revise fig 1f

-page 9, it will be more informative if authors can explain about relationship between cellulose and EPS

-376-377, this scenario has any relation to biofilm?

-386-388, this point is interesting. It will be interesting if authors add one more table/figure to summarize effect of environmental parameters on cellulose synthesis and assembles in cyano, bacteria and plant comparatively

-As a review paper, authors should add 1-2 more table to summary/compare the key properties of cellulose synthesis 

-It will be more interesting if authors can add one small section/paragraph to show currently uses of celluloses in commercial/industries that produced from cyano, bacteria and plants. For example https://ojs.kmutnb.ac.th/index.php/ijst/article/view/7724/pdf_590

Round 2

Reviewer 2 Report

Comments and Suggestions for Authors

accept